# Moisture-induced autonomous surface potential oscillations for energy harvesting

Yu Long[1], Peisheng He[1], Zhichun Shao [1], Zhaoyang Li [2], Han Kim [3], Archie Mingze Yao [4], Yande Peng[1], Renxiao Xu[1], Christine Heera Ahn[1], Seung-Wuk Lee [3], Junwen Zhong[1,2] ✉ & Liwei Lin[1] ✉

A variety of autonomous oscillations in nature such as heartbeats and some biochemical reactions have been widely studied and utilized for applications in the fields of bioscience and engineering. Here, we report a unique phenomenon of moisture-induced electrical potential oscillations on polymers, poly([2-(methacryloyloxy)ethyl] dimethyl-(3-sulfopropyl) ammonium hydroxide-co-acrylic acid), during the diffusion of water molecules. Chemical reactions are modeled by kinetic simulations while system dynamic equations and the stability matrix are analyzed to show the chaotic nature of the system which oscillates with hidden attractors to induce the autonomous surface potential oscillation. Using moisture in the ambient environment as the activation source, this self-excited chemoelectrical reaction could have broad influences and usages in surface-reaction based devices and systems. As a proof-of-concept demonstration, an energy harvester is constructed and achieved the continuous energy production for more than 15,000 seconds with an energy density of 16.8 mJ/cm$^2$. A 2-Volts output voltage has been produced to power a liquid crystal display toward practical applications with five energy harvesters connected in series.

[1] Department of Mechanical Engineering, University of California Berkeley, Berkeley, CA, USA. [2] Department of Electromechanical Engineering and Centre for Artificial Intelligence and Robotics, University of Macau, Macau, SAR, China. [3] Department of Bioengineering, University of California Berkeley, Berkeley, CA, USA. [4] Department of Engineering Mechanics, Tsinghua University, Beijing, China. ✉email: junwenzhong@um.edu.mo; lwlin@berkeley.edu

Electricity has become indispensable in our daily life and many modern devices have been powered by electricity generated from large power plants, while batteries have been the key energy sources for portable devices[1]. In the past decades, green and renewable energy have emerged as alternative power resources, such as solar cells which convert sunlight to electricity[2,3], and piezoelectric or triboelectric energy generators which convert ambient vibration to electrical energy[4,5]. Recently, a new class of energy harvester has been demonstrated by the moisture-electric effect based on the mechanisms of proton transportations[6–15], ion transportations[16–18], and electrokinetic reactions[19–22]. In these systems, the operation time period is intermittent and brief as it is induced by the moisture stimulation in one hydration/dehydration cycle[22]. Since the humidity level does not quickly increase and decrease for the hydration/dehydration cycle in the ambient environment, most reported moisture-electric devices have limited energy outputs. One approach to increase the output energy is to use "protein nanowires"[23] for the continuous electric power generation in the ambient environment. A very different approach is reported here based on the phenomenon of "self-excited proton concentration oscillations" by a carefully designed material system to drastically improve the operation period and electrical energy outputs for the moist-electric generator.

In practice, forced oscillation is a phenomenon to induces repetitive variations by changing the external stimulations, such as force, temperature, light, or electricity input, etc.[24–28]. On the other hand, autonomous oscillations or "self-oscillating" behaviors are the alternating activities without repetitive external stimulations[29–33]. Autonomous oscillations exist widely in living organisms, such as heartbeats, brain waves, and some biochemical reactions[31]. Mimicking and applying these autonomous oscillations in artificial systems have attracted great interest for practical applications. For example, prior efforts have utilized the periodically swelling and de-swelling behavior of polymeric materials caused by the moisture diffusion process induced by sunlight as actuators[34]. Other polymer-based autonomous oscillations have also been demonstrated for applications in artificial muscles, drug-delivery systems, and biosensors, etc.[34–38].

Here, autonomous electrical potential oscillations induced by the moisture diffusion process in poly([2-(methacryloyloxy)ethyl] dimethyl-(3-sulfopropyl) ammonium hydroxide-co-acrylic acid) (P(MEDSAH-co-AA)) are studied. It is found that the surface electrical potential of this polymer system can oscillate continuously upon exposure to moisture to result in the generation of close-circuit alternating current (AC) outputs—this is very different from the previously reported mechanism of direct current (DC) outputs by moisture effects[15]. Experimentally, a porotype energy harvester has been constructed to utilize the autonomous oscillation mechanism with a maximum open-circuit voltage and short-circuit current density of 0.4 V and 1.5 μA/cm², respectively. Furthermore, this device is reusable and the operation time period under one moisture excitation can achieve 15,000 s, with an energy density of 16.8 mJ/cm². The voltage output can be further increased to 2 V by connecting five energy harvesters in series to light up a liquid crystal display (LCD) in a bathroom.

## Results and discussion

**Surface potential oscillation.** One common oscillation example in nature is the concentration of calcium in living cells to assist the transmission of intracellular biological information[39,40]. The calcium-induced $Ca^{2+}$ release (CICR) process is illustrated in Supplementary Fig. 1, where the inositol trisphosphate ($IP_3$) helps the releasing of $Ca^{2+}$ into the cytoplasm from the first pool,

leading to an increase in $Ca^{2+}$ concentration as a positive feedback process. Some similar comparisons can be made for the observed moisture-induced surface potential oscillation phenomenon as the $H^+$ concentration can oscillate with the help of moisture. In the CICR process, some released $Ca^{2+}$ ions are pumped back to the first pool and the second pool of $Ca^{2+}$ also exists in the cytoplasm to release or absorb $Ca^{2+}$ in response to the concentration of the free $Ca^{2+}$. As a result, the $Ca^{2+}$ concentration oscillates continuously. The surface potential of the polymeric material, P(MEDSAH-co-AA) polymer as shown in Fig. 1a, also oscillates as observed in this work. The polymer is synthesized by a free-radical polymerization method[41] with two monomers, [2-(methacryloyloxy)ethyl] dimethyl-(3-sulfopropyl) ammonium hydroxide (MEDSAH) and acrylic acid (AA), initiated by the ammonium persulfate (APS) and polymerized with the detailed process as shown in Supplementary Fig. 2. The ratios of MEDSAH and AA monomers (MEDSAH/AA) in the prototypes are 1/2, 1/3, and 1/4, respectively. To characterize monomer distributions inside the polymer, the energy-dispersive X-ray (EDX) spectroscopy is performed and the N-Kα line is chosen to represent MEDSAH. Supplementary Fig. 3 illustrates that the two monomers are homogeneously distributed in the prototypes of varying MEDSAH/AA ratios. The synthesized polymer is transparent (inset picture in Fig. 1a) and it contains the $N^+$, $-SO_3^-$, and $-COOH$ groups which play key roles in the surface potential oscillation process.

A detailed model is developed to qualitatively explain the electrochemical oscillation process by the related chemical reactions from r1 to r5 in Fig. 1b. The model starts with the release of protons upon the introduction of moisture to activate the functional groups in the polymer as evidenced from the attenuated total reflectance Fourier-transform infrared (ATR-FT-IR) spectroscopy (Supplementary Fig. 4). Specifically, the carboxylic acid ($-COOH$) groups can generate free protons upon exposure to moisture (r1), which increases the polarity and attracts moistures to accelerate the ionization process. This moisture attraction behavior is further validated in Supplementary Fig. 5, where polymers with high $-COOH$ contents can absorb moisture at a faster speed in the hydration process and maintain higher moisture concentrations during the dehydration process as compared with those of polymers with low $-COOH$ contents. As a result, the free proton concentration increases as the positive feedback process. Moistures also help to break the dynamic bonds between $N^+$ and $-SO_3^-$ groups (r2), and the $SO_3^-$ groups can capture protons (r3) to form the sulfonic acid ($-SO_3H$). These two reactions are responsible for the reduction of the proton concentration as the negative feedback processes. Furthermore, protons are transported from the sulfonic acid to the carboxylic acid and $-COOH$ (r4), and the dynamic bonds between the $N^+$ and $-SO_3^-$ groups will re-formulate (r5). The overall processes from r1 to r5 (Supplementary Fig. 6) show that $N^+$, $-SO_3^-$, and $-COOH$ groups complete their individual cycles as moistures diffuse into the P(MEDSAH-co-AA) polymer to assist the self-sustained proton concentration oscillation phenomenon. When there is a moisture concentration gradient inside the polymer, it can induce the anisotropic oscillations of the local proton concentrations to result in the general surface electrical potential oscillations.

Based on these chemical reactions, a kinetic model is established to qualitatively explain the autonomous oscillation mechanism and the exemplified simulation results are shown in Fig. 1c and d. It is found that the proton concentration may oscillate while the relative intensity of the proton concentration decreases gradually over time. The reduction of the proton concentration magnitude over time is also found in the calcium oscillation process in the living cells[40]. Specifically, simulation

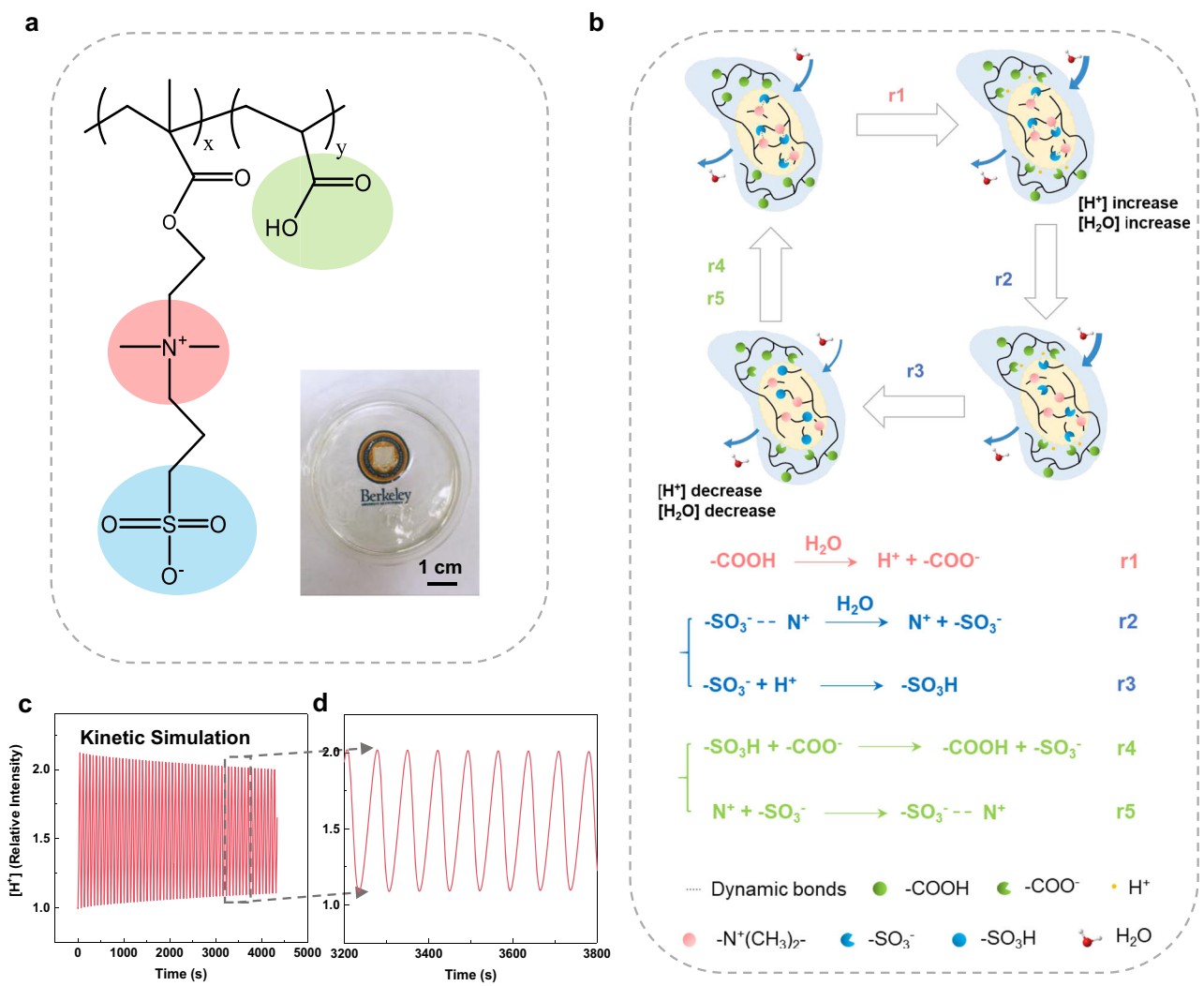

**Fig. 1 Polymer with surface potential oscillation. a** The chemical structure of the as-synthesized poly([2-(methacryloyloxy)ethyl] dimethyl-(3-sulfopropyl) ammonium hydroxide-co-acrylic acid) (P(MEDSAH-co-AA)) polymer with key chemical groups which produce the autonomous oscillation of the surface potential. Inset: an optical photo showing the synthesized and transparent polymer film. **b** Schematic illustrations of the proton transportation process via the functional groups of N+, –SO₃⁻, and –COOH in P(MEDSAH-co-AA) with related chemical reactions: (r1) the proton generation in the positive feedback process, (r2, r3) the proton reduction in the negative feedback processes, and (r4, r5) intermediate interactions. **c** Kinetic simulation results of the H+ concentration versus time from the chemical reactions. **d** Enlarged view of the kinetic simulation results in **c**.

results show that the surface proton intensity decreases to around 87% of its original value after 2000 s and oscillates at about 70 s per cycle for a prototype device by adjusting the parameters to match the experimental observations. By changing the kinetic parameters in the simulation, the oscillation period and intensity degradation will change and these parameters are studied with details in Supporting Explanation 1.

Figure 2a illustrates the surface potential oscillation on the synthesized polymer due to the proton (H+) concentration oscillation. Experimentally, the surface proton concentration oscillation is characterized by using a pH sensor. Figure 2b shows the measured pH vs. time plot in one cycle and Fig. 2c shows the measured voltage vs. time plot of the energy harvester in one cycle. These real-time measurement results demonstrate the proton/voltage oscillation phenomenon. This behavior is believed to come from the chemical potential change as the moisture from the environment diffuses through the polymer structure. Specifically, the chemical potential of water moisture in a high humidity environment is higher than that of absorbed moisture in the polymer. As a result, the change of Gibbs free energy in the

absorbed moisture is converted to electric energy in the chemical potential-based energy harvester[6]. In another test, the KPFM analysis is used to show the surface potential oscillations in a longer, 12-min time period (Supplementary Fig. 7).

The self-excited oscillation is further analyzed in Supporting Explanation 2 by establishing basic equations based on the chemical reactions of r1–r5 and the conservation of matters. Afterward, small perturbations are added to each variable and the system dynamic equations are derived. The stability matrix and phase portraits of the dynamic equations are then analyzed and it is found that the system has a chaotic nature and oscillates with hidden attractors[42]. As a result, this moisture-induced autonomous surface potential oscillation phenomenon is characterized as a chaotic system with no equilibria in favor of extending the operation period. Another resemblance is utilized based on the biological CICR process in Supporting Explanation 3. Specifically, the IP₃ in the CICR process plays a role similar to H₂O to start the proton concentration oscillation process. The first pool in the CICR process is in analogy to the –COOH group for the positive feedback process, and the second pool in the CICR

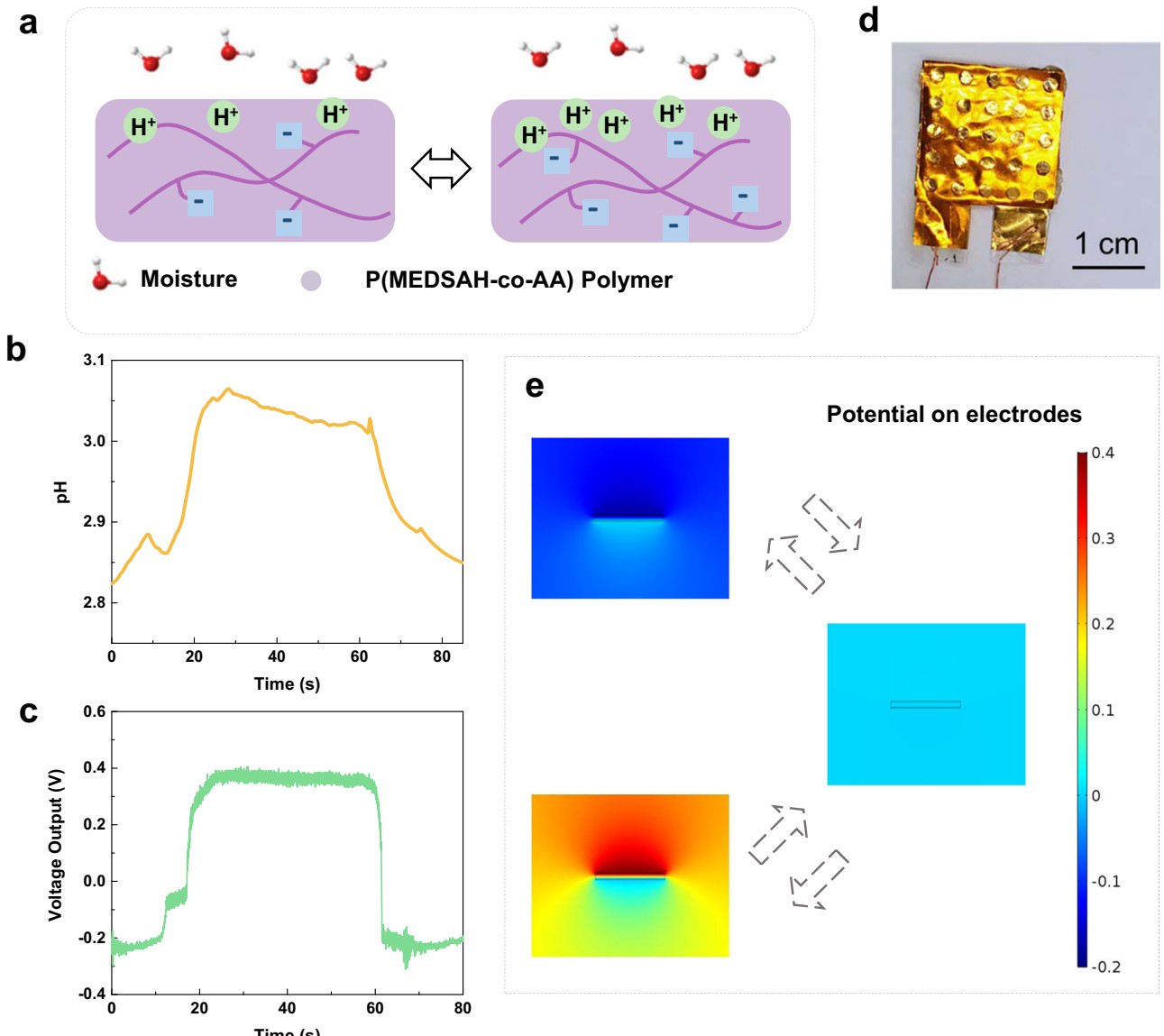

**Fig. 2 Working mechanism verification. a** A schematic diagram showing the surface potential oscillations of poly([2-(methacryloyloxy)ethyl] dimethyl-(3-sulfopropyl) ammonium hydroxide-co-acrylic acid) (P(MEDSAH-co-AA)) polymer. **b** The measured pH vs. time plot for the P(MEDSAH-co-AA) polymer surface in one cycle. **c** Measured voltage vs. time curve in one cycle. **d** An optical photo of a fabricated energy harvester based on P(MEDSAH-co-AA) polymer. **e** COMSOL simulation results showing the surface potential oscillation of an energy harvester versus time.

process is in analogy to and –SO₃⁻ group for the negative feedback process.

The moisture-induced surface potential oscillation is applicable for energy harvesting applications. Previously, several reports have shown moisture-enabled energy harvesters with DC outputs[43–46]. This work demonstrates moisture-induced AC power outputs under a relatively constant moisture level. Figure 2d shows the top view optical photo of a prototype energy harvester made of MEDSAH/AA with a ratio of 1/2 and $2 \times 2 \ cm^2$ in size. The device has one synthesized P(MEDSAH-co-AA) layer sandwiched by two electrodes made Au coated on a polyimide (PI) film and the cross-sectional view is shown in Supplementary Fig. 8. The top electrode has circular holes of 2 mm in diameter to allow direct exposure to moistures. If the P(MEDSAH-co-AA) layer is replaced with a typical copy paper, no visible electrical outputs can be identified (Supplementary Fig. 9) and this result eliminates the possible influence of the electrochemical reactions at the junctions of gold electrodes and copper wires. The surface

potential oscillations on the polymer film can induce the electrostatic potential oscillations on the top electrode, which results in AC outputs as demonstrated in COMSOL simulation results in Fig. 2e and Supplementary Movie 1. The electrostatic stationary simulation is implemented here to illustrate electrostatic potential outputs induced by the oscillation of the surface charges. The simulated output voltage range is between −0.2 and 0.4 V, which is consistent with the experimental results in Fig. 2c.

**Moisture-induced energy harvester with AC outputs.** Experimentally, a specimen has been dried at 30% RH in the ambient environment for 24 h (reducing the inherent moisture to boost the outputs of a chemical potential energy harvester[6]) before a wetting process via an ultrasonic humidifier (Pure Enrichment Inc.). Experimental results show electricity generations in the form of AC outputs after the introduction of 100% RH moisture at 1400 s by the humidifier. It is found that the short-circuit

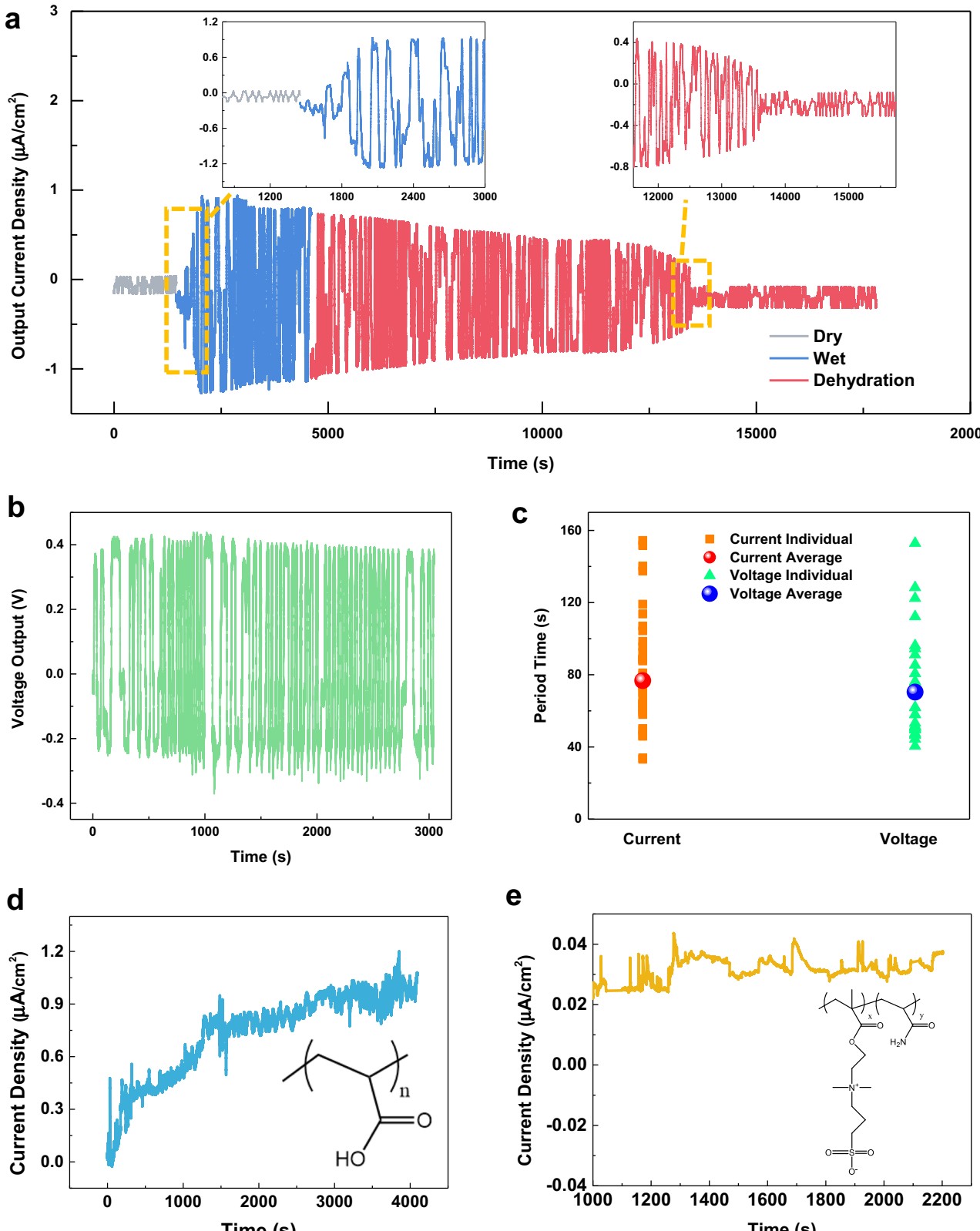

**Fig. 3 Performances of a prototype energy harvester based on surface potential oscillations. a** Short-circuit current density versus time of a prototype energy harvester before the introduction of the moisture (gray color), after the moisture wetting process by a humidifier (blue color), and the dehydration process by removing the humidifier (red color). Insets show enlarged views for the wetting (upper left) and dehydration (upper right) processes. **b** The open-circuit voltage versus time in the moisture wetting process by a humidifier. **c** The testing results of the working cycle period of the open-circuit voltage and short-circuit current. **d** An example of replacing [2-(methacryloyloxy)ethyl] dimethyl-(3-sulfopropyl) ammonium hydroxide (MEDSAH) monomer with poly(acrylic acid) (PAA) to remove the $SO_3^-$ and $N^+$ groups in the polymer, which stops the autonomous oscillations to result in DC electrical outputs. **e** In another example, the acrylic acid (AA) monomer is replaced with acrylamide to remove the –COOH groups, and only DC outputs are also observed.

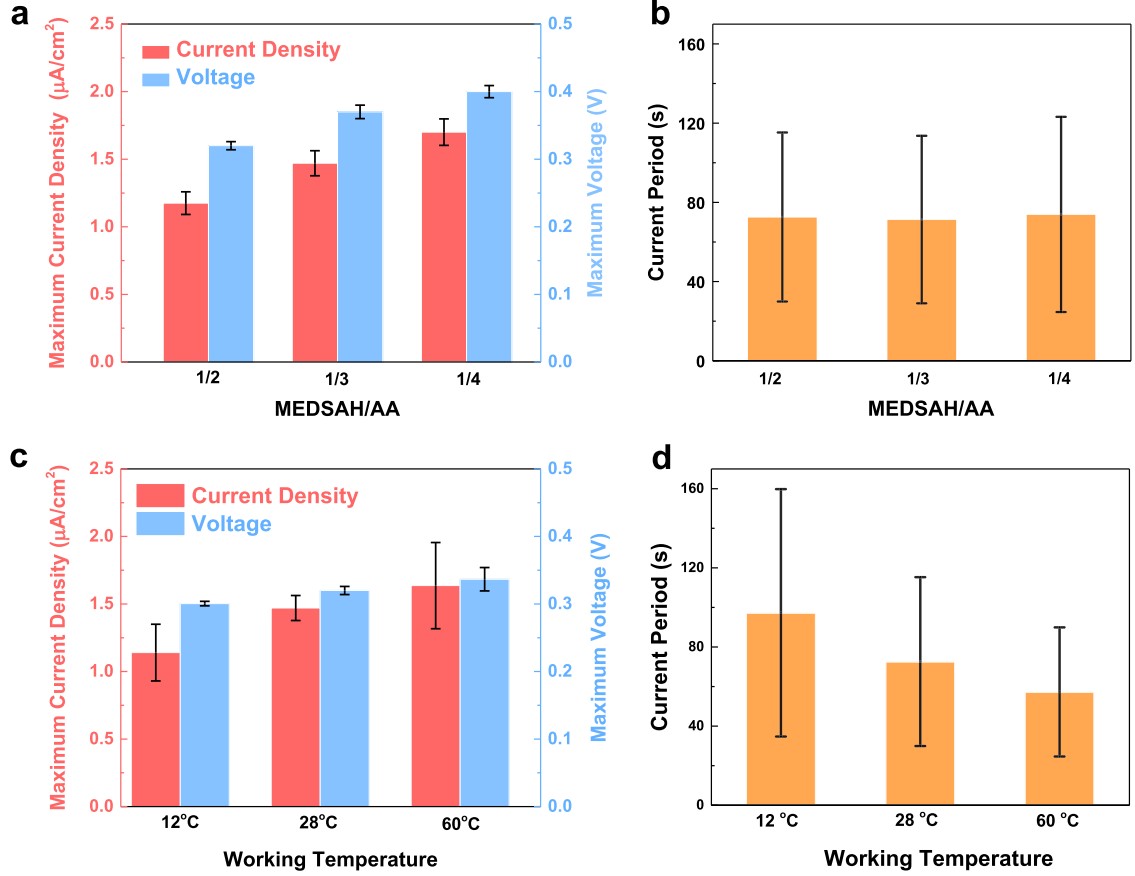

**Fig. 4 Electrical output characterizations for moisture-induced energy harvesters with different monomer ratios and under different temperature environments. a** Maximum short-circuit current density and open-circuit voltage for energy harvesters with the [2-(methacryloyloxy)ethyl] dimethyl-(3-sulfopropyl) ammonium hydroxide (MEDSAH)/acrylic acid (AA) ratios of 1/2, 1/3, and 1/4 at 28 °C. **b** Measured average electrical oscillation output periods of energy harvesters with MEDSAH/AA ratios of 1/2, 1/3, and 1/4, under 28 °C. **c** Maximum short-circuit current density and open-circuit voltage of energy harvesters at operating temperatures of 12, 28, and 60 °C, respectively. **d** Measured average electrical oscillation output periods of energy harvesters at temperatures of 12, 28, and 60 °C, respectively. The error bars show the range of minimum and maximum values.

current density increases initially to a maximum value of around 1 μA/cm² at 2000 s (upper left inset in Fig. 3a). The average oscillation period is about 76.7 s, which is used as the fitting parameter for the kinetic simulation at 70 s (Fig. 1d). The open-circuit voltage versus time result is recorded in Fig. 3b with an average voltage oscillation period of 70.3 s under room temperature and the short-circuit current versus time result is recorded in Fig. 3c. Both tests are independently measured due to their different setups such that minor variations are expected. Due to the chaotic nature of the process (Supporting Explanation 2), simulations cannot precisely predict the experimental results. Nevertheless, main experimental results are summarized as: (1) the average current and voltage oscillation periods in experiments are at 76 and 70 s, respectively; and (2) statistical data show a high number of the operation period around 70 s, while the non-operational period (noises with low-level outputs) is about 150 s (Supplementary Fig. 10).

It is also observed that the current density of the outputs decreases gradually after reaching the peak value. In this test, the humidifier is turned off at 4500 s and the system experiences a natural dehydration process to result in the decrease of the output current (upper right inset in Fig. 3a). At around 13,600 s, a sudden current density drop is observed. Qualitatively, the moisture attraction and evaporation process may reach the equilibrium state over time and there will be no moisture concentration gradient inside the device. As such, the surface potential oscillation process may stop and the energy generation

process ends. Experimentally, it is found that by increasing the external moisture level, the electrical output level also increases (Supplementary Fig. 11), and placing a small droplet of water on the device will also generate electricity with outputs close to those of energy harvesters exposed to the 100% RH ambient condition (Supplementary Fig. 12). In the endurance test under the 100% RH moisture, this energy generation process can last for a long period of time from around 500 to 15,000 s or about 4 h as shown in Supplementary Fig. 13.

Material systems without the key chemical reactions to assist the cyclic process would not be able to induce the self-oscillation process. For example, if the MEDSAH monomer is replaced with poly(acrylic acid) (PAA) to remove –SO₃⁻ and N+ groups, testing results show only DC outputs as shown in Fig. 3d. In another example, –COOH groups are removed by replacing the AA monomer with acrylamide to result in only DC outputs in Fig. 3e. These two counterexamples suggest that the key ingredients and reactions as proposed in the polymer system are indispensable to induce surface potential oscillations for AC outputs of moisture-induced energy harvesters.

**Parameters affecting the outputs and application demonstration.** The electrical outputs of moisture-induced energy harvesters have been studied by changing the compositions of P(MEDSAH-co-AA) polymer and the working temperature. Under a background temperature of 28 °C, the maximum short-

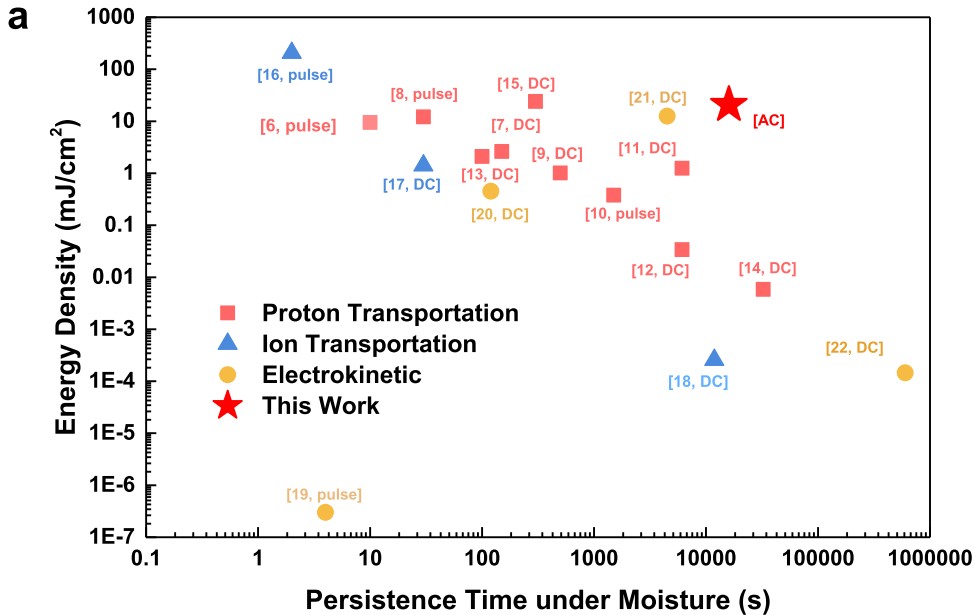

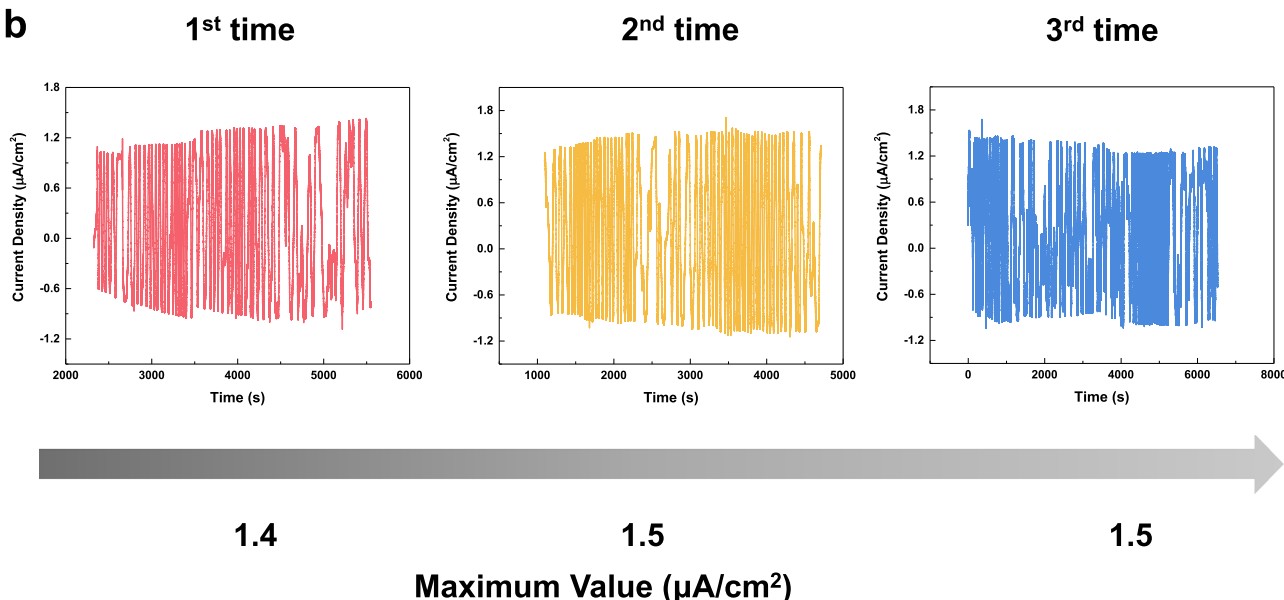

**Fig. 5 The performance comparison between moisture-induced electric generators and the reusability of our prototype energy harvesters. a** Comparing moisture-induced electric generators based on artificial materials in the operation period and energy density. This work has a long persistence time and high-energy-density among moisture-based energy harvesters and these two parameters could be further improved by using other materials based on similar self-oscillation mechanisms. **b** The measured output current versus time results for a prototype energy harvester remains similar in three different independent tests in sequence.

circuit current density and open-circuit voltage versus time of energy harvesters with the MEDSAH/AA ratios of 1/2, 1/3, and 1/4 are shown in Fig. 4a and the current versus time results are shown in Supplementary Fig. 14. In order to reduce the influence of environmental noises, a capacitor (4.7 μF) has been charged by prototype devices with a rectifier and its voltage outputs have been recorded in Supplementary Fig. 15a and b. Other energy harvesters with AC outputs such as the piezoelectric and triboelectric energy harvesters also utilize rectifiers to accumulate the AC electricity in the form of capacitors[47,48]. It is found that by increasing the MEDSAH/AA ratios, the maximum current density and voltage increase from 1.2 μA/cm$^2$ and 0.32 V to 1.7 μA/cm$^2$ and 0.4 V, respectively, due to the increment of the –COOH groups in

the system. The simulation results also predict the increase of oscillation amplitude as the AA composition increases (Supplementary Fig. 16). The average current oscillation period, which is related to the reaction kinetics, is found to maintain at around 72 s for all tested polymers of different compositions as depicted in Fig. 4b. On the other hand, under a MEDSAH/AA ratio of 1/2, the maximum short-circuit current density and open-circuit voltage of a prototype energy harvester under 12, 28, and 60 °C are tested as shown in Fig. 4c (more details in Supplementary Fig. 15 and current outputs in Supplementary Fig. 17). When the temperature is lowered to 12 °C, the maximum current density and voltage slightly decrease to 1.1 μA/cm$^2$ and 0.30 V, while the average oscillation period increases to 97 s (Supplementary Fig. 17a). As the working

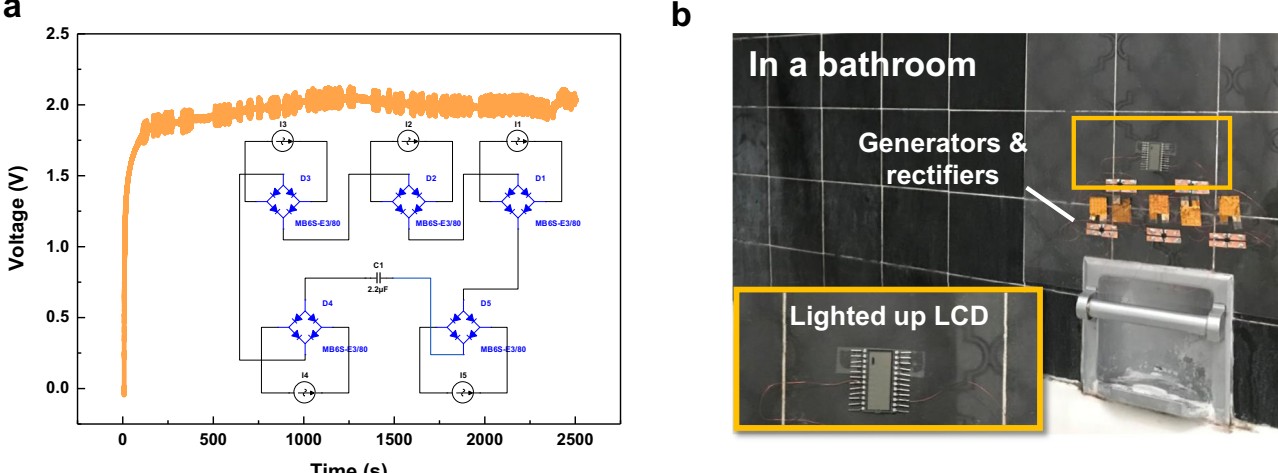

**Fig. 6 Demonstrations of moisture-induced energy harvesters. a** Rectified output voltage vs. time by five devices connected in series. Inset: the electrical circuit. **b** Lighting up a liquid crystal display in a bathroom with moisture from the shower. The inset shows the enlarged view of the system.

temperature increases to 60 °C, the maximum current density and voltage values increase to 1.6 μA/cm² and 0.33 V, and the average oscillation period decreases to 57 s (Supplementary Fig. 17b). Analytically, the reaction speed is increased at a high-temperature environment under energized states and this phenomenon of negative correlation between the temperature and oscillation period is also predicted by simulations results are shown in Supplementary Fig. 18.

The performances of various moisture-enabled energy harvesters using artificial materials are summarized in Fig. 5a. In general, prior works have shown that moisture-induced energy harvesters can generate DC outputs. Here, autonomous surface potential oscillations are utilized to produce AC or voltage outputs and enable long persistence operation time as well as good energy density from prototype devices. For example, testing results show that an operation time of more than 15,000 s (about 4 h), with an output energy density of 16.8 mJ/cm² can be achieved. This long operation time enables sustainable energy outputs under a relatively constant moisture source as the green energy supply. Moreover, the energy harvester is found to be reusable as shown in Fig. 5b, where a maximum current density value at around 1.5 μA/cm² has been found to be maintained for three independent tests.

A simple strategy to increase the output voltage of the energy harvester is to connect devices in series. As shown in Fig. 6a, the outputs of five devices (MEDSAH/AA ratio of 1/4) are rectified and connected in series to reach 2-V, which is high enough to light up an LCD (Supplementary Movie 2). A demonstration toward practical usage is implemented in Fig. 6b by connecting five devices in series on the wall of a bathroom full of moisture during shower time to light up an LCD. This is a direct validation that the moisture-enabled energy harvester can be used in a naturally produced high moisture environments to produce electrical energy.

In summary, a surface potential oscillation phenomenon in moisture-induced P(MEDSAH-co-AA) polymer has been characterized and an AC energy harvester has been demonstrated. Chemical reactions responsible for the autonomous potential oscillation behavior are analyzed and modeled by kinetic simulations to qualitatively characterize this phenomenon. By studying system dynamic equations and the stability matrix, it is found that the system has a chaotic nature and oscillates with hidden attractors to induce autonomous surface potential

oscillation. A prototype energy harvester has a measured maximum short-circuit current density of 1.5 μA/cm² and maximum open-circuit voltage of 0.4 V with a long operation time of up to 15,000 s in a moisture-rich environment to produce an energy density of 16.8 mJ/cm². As a practical application demo, an LCD is successfully lit up by connecting five energy harvesters in series to produce an output voltage of 2 V.

## Methods

**Polymer synthesis process.** All chemicals are bought from Sigma-Aldrich and used without further treatment. First, the monomers and the initiator are weighted according to the designed ratio and solved in DI water with the weight ratio equals 63 wt%. The initiator, APS, is 2 mol% compared with monomers. After fully mixed, the solution is poured into a plastic Petri dish and heated at 50 °C for 24 h. Since the polymers contain dynamic bonds—hydrogen bonds and ionic bonds, their mechanical properties are stable before and after the hydration processes (Supplementary Fig. 19). A rinsing process in DI water is used to get rid of unreacted reactants.

**Device fabrication.** After the polymer synthesis process, an individual specimen is cut into 2 × 2 cm² pieces. A PI film of 20 μm in thickness is sputtered with 100-nm-thick Au as electrodes. The top electrode has 25 mechanically punched holes (diameter of 2 mm) to allow exposure to moistures. Finally, the electrodes are adhered to the polymer structure to complete the device fabrication process.

**Electric signal measurement.** The electrical current is measured via a current amplifier (SR570), and the output voltage is measured by a Data Acquisition system (DAQ, National Instruments).

**pH measurement.** The pH sensor is fabricated via the electrodeposition of aniline as reported previously in the literature[49,50]. For the pH measurement, the top surface of the polymer is first dampened with ~100 μL of water. Then the pH sensor and an Ag/AgCl reference electrode (3 mol/L KCl) are attached to the top surface of the polymer and the open circuit potential between the pH sensor and the reference electrode is measured with an electrochemistry workstation (Gamry Reference 600). The pH values are calculated using a calibration curve obtained from McIlvaine buffer solutions with pH values ranging from 2 to 5. The pH values of McIlvaine buffer solutions are pre-calibrated with a commercial pH meter (Hanna Instruments, HI2210). Supplementary Table 2 shows the measured potential and its corresponding pH value. By linear fitting, it can be found that the reading potential has a good linear relationship with pH with adjacent $R^2$ equals to 0.9942: pH = 14.092 − 0.01545 ∗ Reading potential (Measured data is shown in Supplementary Table 2).

## Characterizations.

(1) EDX tests are performed on a scanning electron microscope (Hitachi 2460 with KEVEX) at 40 keV and the dwelling time is 5 μs.

(2) ATR-FT-IR spectroscopy of dry and wet samples is performed on a Bruker ALPHA Platinum ATR-FT-IR spectrometer equipped with a single reflection diamond ATR module in the ambient atmosphere.

(3) The surface potential is measured using KPFM on MFP-3D atomic force microscopy (AFM; Asylum Research, Santa Barbara, CA) with Ti/Pt-coated AC250TM-R3 (Oxford Instruments) AFM tip with a radius of ~28 nm, spring constant of ~2 N/m, and resonant frequency of ~70 kHz. All measurements are performed at atmospheric pressure, ~20 ºC, ~40% of relative humidity.

## Data availability

All relevant data are available from the corresponding author upon request.

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

## Acknowledgements

The research was performed in part at the Biomolecular Nanotechnology Centre (BNC) lab at the University of California, Berkeley and in part at the Molecular Foundry, Lawrence Berkeley National Laboratory. The Molecular Foundry was supported by the Office of Science, Office of Basic Energy Sciences, of the U.S. Department of Energy under Contract No. DE-AC02-05CH11231. We would like to thank the staff for helps in experiments and Miss Xiaokun Pei for ATR-FT-IR measurement. Special thanks are given to Prof. Jian Qin (Stanford University), Prof. Kuang Yu (Tsinghua-Berkeley Shenzhen Institute) and Prof. Kang Liu (Wuhan University) for discussions and suggestions. The authors would also like to thank Dr. Mallika Bariya (University of California, Berkeley) for the pH sensor; Prof. Juan Pérez-Mercader (Harvard University) for his guidance in the oscillation theory and chaotic behavior study; and Mr. Banghua Zhu (University of California, Berkeley) for his help in oscillation analysis.

## Author contributions

Y.L., J.Z., and L.L. designed the research. Y.L. conducted the synthesis, characterizations, and electric signal measurements. Y.L., P.H. and Y.P. proposed mechanisms, and A.M.Y. established model through simulation. Z.S. conducted COMSOL simulations. H.K. and

S.-W.L. performed KPFM measurements. R.X. provided suggestions on device design and applications. P.H. and C.H.A. performed pH measurements. Z.L. helped on data analysis. Y.L., J.Z., and L.L. discussed the experimental results and drafted the manuscript. J.Z. and L.L. directed this research.

## Competing interests

The authors declare no competing interests.
