## [Peer Review File · Nature Communications]

REVIEWER COMMENTS

Reviewer #1 (Remarks to the Author):

The paper reports a novel phenomenon about the moisture-induced autonomous surface potential oscillation, and introduces its potential applications in energy harvest. The experiments were carefully planned and the results are presented systematically. However, there still are some issues needed to be addressed before the consideration for publication:

(1) Can you present the open-circuit oscillation voltage vs. time curve of one single device? It could be helpful for mechanism understanding.

(2) In this device, the harvested energy comes from the water attraction. In Line 124-125, you mention that “The carboxylic acid (-COOH) groups generate free protons upon the exposure to moisture (r1), which increases the polarity and attracts more moistures to accelerate the ionization process”. Can you provide more evidences to prove the enhanced water attraction? Will this phenomenon increase the harvested energy? Besides, if the water attraction keeps a balance with the water evaporation eventually, will this device continue generating power?

(3) Can you mark and explain where are the 1st pool and 2nd pool in this device? In the schematic diagram of Fig. S6, it seems the 1st pool and 2nd pool are the top and bottom surfaces, respectively. But in Fig. 2, the proton transportations happen between some nearby functional groups. How the proton migration within molecules in nanoscale induces a significant output current in macroscale? Does the current generate because the difference of water concentration between the two surfaces, or the asymmetric orientation of the P(MEDSAH-co-AA) molecules?

(4) As we know, current generation is a process of power consumption. Can you describe the energy conversion process in this device in theory? In some devices, the energy comes from chemical reactions, and the ion flow in the device transform into electron flow by electrochemical reactions, such as the research in ACS Appl. Mater. Interfaces 2020, 12, 21, 24289–24297. Since the reactions in your study is a loop, how the ion flow within the device transforms into electron flow in the external circuit?

Reviewer #2 (Remarks to the Author):

This manuscript reports the copolymer-based AC energy harvester that utilizes the surface potential oscillation behavior by moisture. The moisture was diffused into P(MEDSAH-co-AA) to induce the surface potential oscillation originated in-between the two polymer branches, which results in the AC electricity generation. However, the manuscript requires further clarification on its energy generation mechanism and its electrical potential oscillation to justify their arguments. Therefore, the reviewer feel that the present manuscript is not suitable for publication in Nature Communications. Significant

improvement of the manuscript is required.

Please address the following comments:

1. The design of the copolymer-based AC energy harvesting device is almost same as the previous moist-driven energy harvesters (Energy Environ. Sci. 9, 912–916 (2016)). The previous devices generate both DC forms of voltage and currents by proton or ion transfer phenomena. However, the energy harvester in this work produces the AC form of current and DC form of voltage. This shows that the current generation mechanism and the voltage generation mechanism is clearly different. However, the manuscripts lack information about the mechanism, which makes the reader ambiguous about the energy generation mechanism to understand this work. The author needs to clarify the energy generation mechanism for both voltage and current more clearly compared with previous researches.

1. Author suggested an operation mechanism with the circulation of the protons by scheme r1-r5. However, in the macroscale, no real-time potential oscillation is expected through the given mechanism, as the overall movement of protons is in a dynamic equilibrium state. Thus the real-time proton tracking analysis should be conducted (KPFM analysis has a too broad frequency of data collection). Additionally, the mechanism of the current oscillation needs to be clarified compared to the surface potential oscillation.

2. The kinetic simulation showed in Fig. 1d shows the concentration oscillation over time due to the proton transportation by N^+ , -SO_3^- , COOH groups. However, it is hard to understand the scale of the relative intensity of proton in the y-axis. Besides, the configuration of the macromolecules and the analysis method about the computational simulation need further information to evaluate the results. Lastly, the calculated proton oscillation results are periodical, but the real current measurement lacks periodicity. All of these results need further analysis and clarification.

3. Usage of Fig.2b-e is neither appropriate nor persuasive. The authors interpreted the KPFM data as evidence of potential oscillation. However, the KPFM measurement needs 12 minutes of data accumulation (720 seconds), which is too long to sense the 57~97 second of the current oscillation period.

4. Authors' interpretation of alternative current is suspicious. The author stated $2 \mu\text{A}/\text{cm}^2$ of current was generated at 2000 seconds in Fig.3c. However, it is the summation of the current in the negative direction and positive direction. In the results, the current value of $0.9 \mu\text{A}/\text{cm}^2$ in a positive direction and the current value of $1.1 \mu\text{A}/\text{cm}^2$ in a negative direction was generated, which has $\sim 1 \mu\text{A}/\text{cm}^2$ of alternating current. As a consequence, although the authors emphasized the generation of alternating current, it is hard to accept the significance of the shown performances of the energy harvester. Additionally, the rectifier should be used to utilize the generated power, which seems not to coincide with the concept of paper. Rectifying alternative current implies no evident advantages of alternating current and inducing power drop.

5. In the experimental section, there is no description of washing the synthesized polymer. It is required to add a proper washing procedure for eliminating un-reacted reactants and inhibitors.

6. Authors only discussed the oscillation of current, but the surface potential oscillation that should be read as an alternating voltage is not discussed. In addition, The information of an un-rectified voltage profile should be added and discussed.

7. Fig. 6a should be edited. (white-colored words are not visible.)

Reviewer #3 (Remarks to the Author):

In this manuscript by Long et al., the authors report a unique phenomenon of moisture-induced electrical potential oscillations on P(MEDSAH-co-AA) polymers during the diffusion of water molecules. The new phenomenon of self-excited proton concentration oscillations is found to be used to induce the surface electrical potential oscillates continuously to result in alternating current (AC) electrical outputs. Such a result is quite different from the previously reported mechanism of direct current (DC) induced electricity outputs by moistures. An energy harvester was constructed to demonstrate the continuous energy production for more than 15000 seconds with an energy density of 16.824 mJ/cm². The results reported in this study are interesting. However, major issues on the proposed mechanism and experimental evidence still existed, which is very important to the study. Therefore, at the current stage, this manuscript can not be accept for publication in Nature Communications. However, this manuscript can be re-considered after the author addressing these issues.

Major issues:

1. It is noticed that the AC outputs experiments were carried out at 100% RH in this work, so how sensitive is the P(MEDSAH-co-AA) polymer to humidity? If it could work under normal humidity, for example, 60% RH, or other lower humidity environment conditions? The electrical output performance under different RH should be provided.

2. The author described that “a specimen has been dried at 30% RH for 24 hours before a wetting process via an ultrasonic humidifier”. Does this mean that the inherent water molecules absorbed by P(MEDSAH-co-AA) polymer would weaken the performance of alternating current? Can we assume that the P(MEDSAH-co-AA) polymer placed in a normal ambient humidity without drying pretreatment does not have the phenomenon of moisture-induced electrical potential oscillations?

3. The authors used the kinetic simulation method to clarify the electrochemical oscillation process. I suggest that more experiments need to be designed to prove it, more experimental data need to be provided to support the mechanism in this work.

4. Compared with moisture-induced autonomous surface potential oscillations for energy harvesting, we believe that water-induced autonomous surface potential oscillations for energy harvesting are more accurate according to the description of this work. How about dropping a small amount of water to the P(MEDSAH-co-AA) polymer directly to observe the performance of electricity?

5. I noticed that the Au electrodes and wires were soldered together with tin without encapsulation as shown in Movie S2. Is there any redox reaction at the junction when being exposed to 100% RH?

6. Comparisons of using PAA only should be provided in the manuscript.

7. It is noted the P(MEDSAH-co-AA) polymers were used without crosslinking. Therefore, the stability of the materials or the devices should be considered.

List of Changes and Response to Reviewer Comments

Reviewer # 1

General Comment

The paper reports a novel phenomenon about the moisture-induced autonomous surface potential oscillation and introduces its potential applications in energy harvest. The experiments were carefully planned, and the results are presented systematically. However, there still are some issues needed to be addressed before the consideration for publication:

Response: We thank the reviewer for the positive comments. We have modified the manuscript per the referee's constructive comments and suggestions.

Comment 1: Can you present the open-circuit oscillation voltage vs. time curve of one single device? It could be helpful for mechanism understanding.

Response: The measured open-circuit voltage versus time plot is added in **Fig. 3b** and the average oscillation period is 70.3 seconds, which is closed to that of the short-circuit current result at 76.7 seconds (**Fig. 3c**). The open-circuit voltage and short-circuit current tests were conducted independently.

Fig. 3. (b) The open-circuit voltage versus time in the moisture wetting process by a humidifier. (c) The testing results of the working cycle period of the open-circuit voltage and short-circuit current.

Comment 2: In this device, the harvested energy comes from the water attraction. In Line 124-125, you mention that “The carboxylic acid (-COOH) groups generate free protons upon the exposure to moisture (r1), which increases the polarity and attracts more moistures to accelerate the ionization process”. Can you provide more evidences to prove the enhanced water attraction? Will this phenomenon increase the harvested energy? Besides, if the water attraction keeps a balance with the water evaporation eventually, will this device continue generating power?

Response: The moisture contents versus time plots during the hydration and dehydration process for polymers of different compositions are added in **Figs. S5a and S5b**, respectively. It is observed that polymers with high carboxylic acid group (-COOH) can absorb more moisture at a faster speed in the hydration process and maintain higher moisture concentrations in the dehydration process. This phenomenon does increase the harvested energy as depicted in **Fig. 4a**. However, at the equilibrium state, there will be no moisture concentration gradient and the energy harvesting process will stop.

Fig. S5. The water content of polymers with different compositions during: (a) the hydration and (b) the dehydration process. It can be found that the P(MEDSAH-co-AA) polymer containing more -COOH can absorb more moisture at a faster speed in (a) and maintain higher moisture concentrations in (b).

Fig. 4a. Maximum short-circuit current density and open-circuit voltage for energy harvesters with the MEDSAH/AA ratios of 1/2, 1/3, and 1/4, at 28°C.

Comment 3: Can you mark and explain where are the 1st pool and 2nd pool in this device? In the schematic diagram of Fig. S6, it seems the 1st pool and 2nd pool are the top and bottom surfaces, respectively. But in Fig. 2, the proton transportations happen between some nearby functional groups. How the proton migration within molecules in nanoscale induces a significant output current in macroscale? Does the current generate because the difference of water concentration between the two surfaces, or the asymmetric orientation of the P(MEDSAH-co-AA) molecules?

Response: The resemblance of the biological Calcium-induced Ca^{2+} release (CICR) process to the current work is utilized to help further explain the phenomenon while the one-to-one matching may not be fully established. As such, we have moved the comparisons to **Supporting Explanation 3** with an illustrating Table. Specifically, the IP_3 in the CICR process plays a role similar to H_2O to start the proton concentration oscillation process. The 1st pool in the CICR process is in analogy to the $-\text{COOH}$ group for the positive feedback process, and the 2nd pool in the CICR process is in analogy to and $-\text{SO}_3^-$ group for the negative feedback process. The self-excited oscillation is further analyzed in **Supporting Explanation 2** by establishing basic equations based on chemical reactions of r1 to r5 and the conservation of matters. Afterwards, small perturbations are added to each variable and the system dynamic equations

are derived. The stability matrix and phase portraits (**Fig. S22**) of the dynamic equations are then analyzed and it is found that the system has the chaotic nature and oscillates with hidden attractors. As a result, this moisture-induced autonomous surface potential oscillation phenomenon is characterized as a chaotic system with no equilibria in favor of extending the operation period.

Comparison between this work and Ca^{2+} oscillation

Ca^{2+} oscillation	This work	Similarity
IP_3	H_2O	Start the oscillation
Ca^{2+}	H^+	Oscillated signal
1 st pool	$-\text{COOH}$	In positive feedback(s)
2 nd pool	$-\text{SO}_3^-$	In negative feedback(s)

Fig. S22. The phase portraits of the system, showing the oscillating and chaotic behaviors. The

above six figures are projections on different concentration coordinate systems in the four-dimensional space. The projection planes are (a) H^+ and $-\text{SO}_3^-$; (b) H^+ and $-\text{SO}_3^-\text{N}^+$; (c) H^+ and H_2O ; (d) H_2O and $-\text{SO}_3^-$; (e) H_2O and $-\text{SO}_3^-\text{N}^+$; (f) $-\text{SO}_3^-\text{N}^+$ and $-\text{SO}_3^-$.

Comment 4: As we know, current generation is a process of power consumption. Can you describe the energy conversion process in this device in theory? In some devices, the energy comes from chemical reactions, and the ion flow in the device transform into electron flow by electrochemical reactions, such as the research in ACS Appl. Mater. Interfaces 2020, 12, 21, 24289–24297. Since the reactions in your study is a loop, how the ion flow within the device transforms into electron flow in the external circuit?

Response: We have added the surface proton concentration oscillation measurements by using a pH sensor. **Fig. 2b** shows the measured pH vs. time plot in one cycle and **Fig. 2c** shows the measured voltage vs. time plot of the energy harvester in one cycle. These real-time measurement results demonstrate the proton/voltage oscillation phenomenon. This behavior is believed to come from the chemical potential change as the moisture from the environment diffuses through the polymer structure. Specifically, the chemical potential of water moisture in a high humidity environment is higher than that of absorbed moisture in the polymer. As a result, the change of Gibbs free energy in the absorbed moisture is converted to electric energy in the chemical potential-based energy harvester¹ instead of galvanic corrosion and water decomposition².

Fig. 2 (b) The measured pH vs. time plot for the P(MEDSAH-co-AA) polymer surface in one cycle. (c) Measured voltage vs. time curve in one cycle.

1. Zhao, F., Liang, Y., Cheng, H., Jiang, L. & Qu, L. Highly efficient moisture-enabled electricity generation from graphene oxide frameworks. *Energy Environ. Sci.* **9**, 912–916 (2016).
2. Feng, J. *et al.* High-Performance Magnesium–Carbon Nanofiber Hygroelectric Generator Based on Interface-Mediation-Enhanced Capacitive Discharging Effect. *ACS Appl. Mater. Interfaces* **12**, 24289–24297 (2020).

Reviewer: #2

General Comment

This manuscript reports the copolymer-based AC energy harvester that utilizes the surface potential oscillation behavior by moisture. The moisture was diffused into P(MEDSAH-co-AA) to induce the surface potential oscillation originated in-between the two polymer branches, which results in the AC electricity generation. However, the manuscript requires further clarification on its energy generation mechanism and its electrical potential oscillation to justify their arguments. Therefore, the reviewer feel that the present manuscript is not suitable for publication in Nature Communications. Significant improvement of the manuscript is required.

Response: We thank the reviewer for reviewing our manuscript carefully and give us helpful and detailed comments. We have modified the manuscript per the referee's constructive comments and suggestions.

Comment 1: The design of the copolymer-based AC energy harvesting device is almost same as the previous moist-driven energy harvesters (*Energy Environ. Sci.* **9**, 912–916 (2016)). The previous devices generate both DC forms of voltage and currents by proton or ion transfer phenomena. However, the energy harvester in this work produces the AC form of current and DC form of voltage. This shows that the current generation mechanism and the voltage generation mechanism is clearly different. However, the manuscripts lack information about the mechanism, which makes the reader ambiguous about the energy generation mechanism to understand this work. The author needs to clarify the energy generation mechanism for both voltage and current more clearly compared with previous research.

Response: Our polymer system is very different than the one used in the previous moist-driven

energy harvesters.¹ In fact, both the output voltage and current generated in this work are in AC forms. In the initial submission, we used a rectifier to convert the AC voltage output to DC voltage. In this revision, we directly measured the open-circuit AC voltage versus time, as shown in **Fig. 3b** and the average oscillation period is 70.3 seconds, which is close to that of the short-circuit current result at 76.7 seconds (**Fig. 3c**). Our moisture-induced energy harvester belongs to a chemical potential energy harvester.¹ The mechanism starts from the chemical potential of water moisture in a high humidity environment higher than that of absorbed moisture in the P(MEDSAH-co-AA) polymer. As a result, the change of Gibbs free energy in the absorbed moisture is converted to electric energy in the chemical potential-based energy harvester instead of galvanic corrosion or water decomposition². Different from the previous research,¹ the surface potential oscillations on the P(MEDSAH-co-AA) polymer film can induce the electrostatic potential oscillations on the top electrode, which results in AC outputs as demonstrated in COMSOL simulation results in **Fig. 2e** and **Movie S1**. The electrostatic stationary simulation is implemented here to illustrate electrostatic potential outputs induced by the oscillation of the surface charges. The simulated output voltage range is between -0.2 to 0.4 V, which is consistent with the experimental results in **Fig. 3b**.

1. Zhao, F., Liang, Y., Cheng, H., Jiang, L. & Qu, L. Highly efficient moisture-enabled electricity generation from graphene oxide frameworks. *Energy Environ. Sci.* **9**, 912–916 (2016).
2. Feng, J. *et al.* High-Performance Magnesium–Carbon Nanofiber Hygroelectric Generator Based on Interface-Mediation-Enhanced Capacitive Discharging Effect. *ACS Appl. Mater. Interfaces* **12**, 24289–24297 (2020).

Fig. 3. (b) The open-circuit voltage versus time in the moisture wetting process by a humidifier. (c) The testing results of the working cycle period of the open-circuit voltage and short-circuit current.

Fig. 2e. COMSOL simulation results showing the surface potential oscillation of an energy harvester versus time.

Comment 2: Author suggested an operation mechanism with the circulation of the protons by scheme r1-r5. However, in the macroscale, no real-time potential oscillation is expected through the given mechanism, as the overall movement of protons is in a dynamic equilibrium

state. Thus the real-time proton tracking analysis should be conducted (KPFM analysis has a too broad frequency of data collection). Additionally, the mechanism of the current oscillation needs to be clarified compared to the surface potential oscillation.

Response: We agree with the reviewer that the KPFM result is too broad frequency of data collection and has moved it to **Supplementary Materials**. In order to further analyze the operation mechanisms, we have added results from the analyses of stability matrix and phase portraits for better explanations (**Supporting Explanation 2**)³ by establishing basic equations based on chemical reactions of r1 to r5 and the conservation of matters. Afterwards, small perturbations are added to each variable and the system dynamic equations are derived. The stability matrix and phase portraits (**Fig. S22**) of the dynamic equations are then analyzed and it is found that the system has the chaotic nature and oscillates with hidden attractors. As a result, this moisture-induced autonomous surface potential oscillation phenomenon is characterized as a chaotic system with no equilibria in favor of extending the operation period.

3. Panahi, S., Pham, V.-T., Rajagopal, K., Boubaker, O. & Jafari, S. A New Four-Dimensional Chaotic System With No Equilibrium Point. in *Recent Advances in Chaotic Systems and Synchronization* 63–76 (Elsevier, 2019). doi:10.1016/b978-0-12-815838-8.00004-2.

Fig. S22. The phase portraits of the system, showing the oscillating and chaotic behaviors. The above six figures are projections on different concentration coordinate systems in the four-dimensional space. The projection planes are (a) H^+ and $-SO_3^-$; (b) H^+ and $-SO_3^-N^+$; (c) H^+ and H_2O ; (d) H_2O and $-SO_3^-$; (e) H_2O and $-SO_3^-N^+$; (f) $-SO_3^-N^+$ and $-SO_3^-$.

Comment 3: The kinetic simulation showed in Fig. 1d shows the concentration oscillation over time due to the proton transportation by N^+ , $-SO_3^-$, $COOH$ groups. However, it is hard to understand the scale of the relative intensity of proton in the y-axis. Besides, the configuration of the macromolecules and the analysis method about the computational simulation need further information to evaluate the results. Lastly, the calculated proton oscillation results are periodical, but the real current measurement lacks periodicity. All of these results need further analysis and clarification.

Response: In order to simulate the system, we use upwind explicit differential scheme to solve the partial differential equations in the kinetic simulation (**Supporting Explanations 1**), and

the results have been normalized in **Fig. 1c** with respect to the initial proton concentration which corresponds to the relative intensity of protons. In order to further analyze the operation mechanisms, we have added results from the analyses of stability matrix and phase portraits for better explanations (See detailed responses in **Comment 2**). Due to the chaotic nature of the process, kinetic simulations cannot precisely predict the experimental results. Nevertheless, main experimental results are summarized as: (1) the average current and voltage oscillation periods in experiments are at 76 and 70 seconds, respectively; and (2) statistical data show a high number of the operation period around 70 seconds, while the non-operational period (noises with low level outputs) is about 150 seconds in the added figure in **Fig. S10**. These provide better analyses and explanations for this work.

Fig. S10. The histogram of energy generation cycle period of a prototype energy harvester: (a) energy generation period; and (b) low electrical output periods (noises).

Comment 4: Usage of Fig.2b-e is neither appropriate nor persuasive. The authors interpreted the KPFM data as evidence of potential oscillation. However, the KPFM measurement needs 12 minutes of data accumulation (720 seconds), which is too long to sense the 57~97 second of the current oscillation period.

Response: We agree with the reviewer that the KPFM result is too broad frequency of data collection and has moved it to **Supplementary Materials**. Furthermore, we added new results by probing the pH variation on the P(MEDSAH-co-AA) polymer surface in **Fig. 2b**. The pH vs. time curves has similar waveform shape and period time with voltage vs. time curves of the

energy harvester based on the P(MEDSAH-co-AA) polymer (**Fig. 2c**). These real-time results help illustrating the proton concentration oscillations to induce the AC outputs.

Fig. 2 (b) The measured pH vs. time plot for the P(MEDSAH-co-AA) polymer surface in one cycle. (c) Measured voltage vs. time curve in one cycle.

Comment 5: Authors' interpretation of alternative current is suspicious. The author stated $2 \mu\text{A}/\text{cm}^2$ of current was generated at 2000 seconds in Fig.3c. However, it is the summation of the current in the negative direction and positive direction. In the results, the current value of $0.9 \mu\text{A}/\text{cm}^2$ in a positive direction and the current value of $1.1 \mu\text{A}/\text{cm}^2$ in a negative direction was generated, which has $\sim 1 \mu\text{A}/\text{cm}^2$ of alternating current. As a consequence, although the authors emphasized the generation of alternating current, it is hard to accept the significance of the shown performances of the energy harvester. Additionally, the rectifier should be used to utilize the generated power, which seems not to coincide with the concept of paper. Rectifying alternative current implies no evident advantages of alternating current and inducing power drop.

Response: The maximum current density comment from the reviewer is correct. We have changed our expression to maximum values in the revised **Figs. 4a and 4c**. On the other hand, the alternating current generation from other types of energy harvesters is common such as piezoelectric and triboelectric energy harvesters. For application demonstration, many other works also use rectifiers to accumulate the AC electricity in capacitors.^{4,5}

4. Hu Y, Zhang Y, Xu C, Lin L, Snyder RL, Wang ZL. Self-powered system with wireless data transmission. *Nano Lett.* **11**, 2572-2577 (2011).
5. Niu S, Wang X, Yi F, Zhou YS, Wang ZL. A universal self-charging system driven by random biomechanical energy for sustainable operation of mobile electronics. *Nat. Commun.* **6**, 8975 (2015).

Fig. 4 (a) Maximum short-circuit current density and open-circuit voltage for energy harvesters with the MEDSAH/AA ratios of 1/2, 1/3, and 1/4, at 28°C. (c) Maximum short-circuit current density and open-circuit voltage of energy harvesters under temperatures of 12, 28, and 60°C, respectively.

Comment 6: In the experimental section, there is no description of washing the synthesized polymer. It is required to add a proper washing procedure for eliminating un-reacted reactants and inhibitors.

Response: The polymer system has an ultra-high molecular weight and contains a large amount of ionic bonds and hydrogen bonds which can serve as crosslinkers. We did not use the common liquid extraction method to wash the polymer but we did clean the polymer by rinsing it with DI water and this method is added in the revised manuscript.

Comment 7: Authors only discussed the oscillation of current, but the surface potential oscillation that should be read as an alternating voltage is not discussed. In addition, the information of an un-rectified voltage profile should be added and discussed.

Response: The measured open-circuit voltage versus time plot is added in **Fig. 3b**. Please also see other detailed responses in **Comment 1**.

Comment 8: Fig. 6a should be edited. (white-colored words are not visible.)

Response: We have addressed this problem in the revised manuscript.

Reviewer # 3

General Comment

In this manuscript by Long et al., the authors report a unique phenomenon of moisture-induced electrical potential oscillations on P(MEDSAH-co-AA) polymers during the diffusion of water molecules. The new phenomenon of self-excited proton concentration oscillations is found to be used to induce the surface electrical potential oscillates continuously to result in alternating current (AC) electrical outputs. Such a result is quite different from the previously reported mechanism of direct current (DC) induced electricity outputs by moistures. An energy harvester was constructed to demonstrate the continuous energy production for more than 15000 seconds with an energy density of 16.824 mJ/cm². The results reported in this study are interesting. However, major issues on the proposed mechanism and experimental evidence still existed, which is very important to the study. Therefore, at the current stage, this manuscript can not be accept for publication in Nature Communications. However, this manuscript can be re-considered after the author addressing these issues.

Response: We thank the reviewer for reviewing our manuscript carefully and give us helpful and detailed comments. We have modified the manuscript per the referee's constructive comments and suggestions.

Comment 1: It is noticed that the AC outputs experiments were carried out at 100% RH in this work, so how sensitive is the P(MEDSAH-co-AA) polymer to humidity? If it could work under normal humidity, for example, 60% RH, or other lower humidity environment conditions? The electrical output performance under different RH should be provided.

Response: The electrical output performances under different RH are added as shown in **Fig. S11**. It is found that low humidity levels will lead to low current density outputs.

Fig. S11. The measured output current density versus time of the prototype energy harvester under various humidity levels of (a) 59% RH, (b) 80% RH, and (c) 90% RH. (d) Average maximum current density versus relative humidity.

Comment 2: The author described that “a specimen has been dried at 30% RH for 24 hours before a wetting process via an ultrasonic humidifier”. Does this mean that the inherent water molecules absorbed by P(MEDSAH-co-AA) polymer would weaken the performance of alternating current? Can we assume that the P(MEDSAH-co-AA) polymer placed in a normal ambient humidity without drying pretreatment does not have the phenomenon of moisture-induced electrical potential oscillations?

Response: The initial inherent moisture will weaken the electrical outputs as this energy harvester is based on the chemical potential changes from the moisture with a potential in the high humidity environment to a low potential when the moisture is absorbed by the polymer.

Our real-time measurement results demonstrate the proton/voltage oscillation phenomenon. This behavior is believed to come from the chemical potential change as the moisture from the environment diffuses through the polymer structure. Specifically, the chemical potential of water moisture in a high humidity environment is higher than that of absorbed moisture in the polymer. As a result, the change of Gibbs free energy in the absorbed moisture is converted to electric energy in the chemical potential-based energy harvester.¹ The input energy is the chemical potential change of water from high humidity environment, μ_h , to low humidity polymer, μ_l . In this case, μ_h is a constant, as the outside concentration of moisture, C_0 , is a constant; μ_l is also a constant which can be derived from the water concentration absorbed in the polymer, ΔC . The change of Gibbs energy:

$$\Delta G = \mu_h - \mu_l \approx RT \ln\left(\frac{C_0}{C_0 - \Delta C}\right)$$

Without sufficient drying pretreatment, ΔC could be small and without a high environmental humidity, C_0 could be small.

1. Zhao, F., Liang, Y., Cheng, H., Jiang, L. & Qu, L. Highly efficient moisture-enabled electricity generation from graphene oxide frameworks. *Energy Environ. Sci.* **9**, 912–916 (2016).

Comment 3: The authors used the kinetic simulation method to clarify the electrochemical oscillation process. I suggest that more experiments need to be designed to prove it, more experimental data need to be provided to support the mechanism in this work.

Response: We have added more theoretical analyses to clarify the electrochemical oscillation process by using the stability matrix and phase portraits for the chaotic system (**Supporting Explanation 2 and Fig. S22**). We have also added more experiments to clarify and support the proposed mechanism: (1) **Figs. S5a and S5b** are added for measured water content of polymers with different compositions during: (a) the hydration and (b) the dehydration process to show that the polymer containing more -COOH can absorb more moisture at a faster speed in (a) and maintain higher moisture concentrations in (b); (2) **Figs. 2d and 2c** are added for measured pH vs. time plot for the P(MEDSAH-co-AA) polymer surface in one cycle and measured voltage vs. time curve in one cycle of a fabricated energy harvester with similar waveform shape and

period time; (3) **Figs. 3d and Fig. 3e** are outputs of the polymers containing only one kind of functional groups with only DC outputs; and (4) **Fig. S9** is added to show that the AC outputs come from the proton oscillation instead of the electrochemical reactions at the junction of gold electrodes and copper wires.

Fig. S5. The water content of polymers with different compositions during: (a) the hydration and (b) the dehydration process. It can be found that the P(MEDSAH-co-AA) polymer containing more -COOH can absorb more moisture at a faster speed in (a) and maintain higher moisture concentrations in (b).

Fig. 2 (b) The measured pH vs. time plot for the P(MEDSAH-co-AA) polymer surface in one cycle. (c) Measured voltage vs. time curve in one cycle.

Fig. 3. (d) An example of replacing MEDSAH monomer with PAA to remove the SO_3^- and N^+ groups in the polymer, which stops the autonomous oscillations to result in DC electrical outputs. (e) In another example, the AA monomer is replaced with acrylamide to remove the $-\text{COOH}$ groups and only DC outputs are observed.

Fig. S9. The (a) close-circuit current and (b) open-circuit voltage vs. time results of a prototype device by using a copy paper instead of the P(MEDSAH-co-AA) polymer. No visible electrical outputs are identified.

Comment 4: Compared with moisture-induced autonomous surface potential oscillations for energy harvesting, we believe that water-induced autonomous surface potential oscillations for energy harvesting are more accurate according to the description of this work. How about dropping a small amount of water to the P(MEDSAH-co-AA) polymer directly to observe the performance of electricity?

Response: We have added experimental results by placing water droplet on the device to generate electricity and the performances are close to those of energy harvesters exposed to the condition of 100% RH moisture in **Fig. S12**. Since moisture under different RH also generates electricity as shown in **Fig. S11**, we prefer using “moisture” instead of “water” for our device.

Fig. S12. The output current vs. time plot by placing a water droplet on the device.

Comment 5: I noticed that the Au electrodes and wires were soldered together with tin without encapsulation as shown in Movie S2. Is there any redox reaction at the junction when being exposed to 100% RH?

Response: We added one more experiment by replacing the polymer layer with a commercial copy paper and visible electrical outputs were identified as shown in **Fig. S9** to illustrate no redox reaction at the junctions.

Comment 6: Comparisons of using PAA only should be provided in the manuscript.

Response: The comparisons of only using PAA is provided in the manuscript in **Fig. 3d** as this device only generates DC electricity.

Comment 7: It is noted the P(MEDSAH-co-AA) polymers were used without crosslinking. Therefore, the stability of the materials or the devices should be considered.

Response: Since the polymers contain dynamic bonds – hydrogen bonds and ionic bonds, their

mechanical properties are stable before and after the hydration processes (**Fig. S19**). A rinsing process in DI water is used to get rid of unreacted reactants.

Fig. S19. The measured stress-strain curves of the polymer (a) before and (b) after the hydration process. The calculated Young's modulus are 30.6 MPa and 66.8 kPa before and after hydration, respectively.

REVIEWERS' COMMENTS

Reviewer #1 (Remarks to the Author):

The authors provided some useful information and added additional experiment data and explanations to support their conclusions. But there still are questions needed to be answered properly:

(1) The measured open-circuit voltage versus time plot is added. An open-circuit voltage of ~ 0.4 V is comparable to the previously reported results. The AC voltage and current signals are interesting but not necessary for the promotion of voltage and current output. What are the advantages of AC signals compared to DC signals?

(2) The author still did not explain how the proton migration within molecules in nanoscale induces a significant output current in macroscale, in Comment 3. In your device, the distribution of P(MEDSAH-co-AA) polymer is isotropic. To my understanding, power generation is caused by the water concentration difference between the surface and the middle. The 1st pool is the -COOH on the surface, and the 2nd pool is the -SO₃⁻ in the middle. Please clarify this.

Reviewer #2 (Remarks to the Author):

I recommend the publication of revised manuscript. The authors addressed well the comments in responses.

Reviewer #3 (Remarks to the Author):

In this revised manuscript, the authors have made proper modification according to the reviewers' comments. I believe that this manuscript in the current form can be accepted for publication in Nature Communications.

List of Changes and Response to Reviewer Comments

Reviewer # 1

General Comment

The authors provided some useful information and added additional experiment data and explanations to support their conclusions. But there still are questions needed to be answered properly:

Response: We thank the reviewer for positive comments. We have modified the manuscript per the referee's constructive comments.

Comment 1: The measured open-circuit voltage versus time plot is added. An open-circuit voltage of ~ 0.4 V is comparable to the previously reported results. The AC voltage and current signals are interesting but not necessary for the promotion of voltage and current output. What are the advantages of AC signals compared to DC signals?

Response: The reviewer is correct that AC voltage and current signals are interesting but not necessary for the promotion of voltage and current outputs. The important parameter of an energy harvester is the overall energy outputs. In this regard, we have summarized the published works in the field of moisture-induced energy harvesters in Fig. 5a. Autonomous surface potential oscillations are utilized to produce alternating current (AC) or voltage outputs and enable long persistence operation time as well as good energy density from prototype devices, while longer operation time and higher energy density devices/mechanisms should be future explored.

Fig. 5a. Comparison moisture-induced electric generators based on artificial materials in the operation period and energy density. This work has the long persistence time and high energy density among moisture-based energy harvesters, and these two parameters could be further improved by using other materials based on similar self-oscillation mechanisms.

Comment 2: The author still did not explain how the proton migration within molecules in nanoscale induces a significant output current in macroscale, in Comment 3. In your device, the distribution of P(MEDSAH-co-AA) polymer is isotropic. To my understanding, power generation is caused by the water concentration difference between the surface and the middle. The 1st pool is the -COOH on the surface, and the 2nd pool is the -SO³⁻ in the middle. Please clarify this.

Response: The reviewer is correct that the distribution of P(MEDSAH-co-AA) polymer is isotropic, and the energy generation stops when the moisture concentration is fully saturated inside the polymer (no concentration gradient). In general, our energy harvester is based on the chemical potential changes. It is observed that the chemical potential of moisture in a high humidity environment is larger than that in the polymer and the Gibbs free energy change from the absorbed moisture is transferred to the electrical energy. Specifically, the chemical potential of moisture from the high humidity environment, μ_h , is higher than the chemical potential of moisture in the low humidity polymer, μ_l .

The change of Gibbs energy is expressed as:

$$\Delta G = \mu_h - \mu_l \approx RT \ln\left(\frac{C_0}{C_0 - \Delta C}\right)$$

where C_0 is the moisture concentration in the high humidity environment and ΔC is the moisture concentration absorbed by the polymer. The 1st pool is the -COOH and the 2nd pool is the -SO₃⁻ in the same location of the polymer. When there is moisture concentration gradient inside the polymer, it can induce the anisotropic oscillations of the local proton concentrations to result in the general surface electrical potential oscillations and AC outputs.

Reviewer: #2

General Comment

I recommend the publication of revised manuscript. The authors addressed well the comments in responses.

Response: We thank the reviewer for recommending the publication of this paper.

Reviewer # 3

General Comment

In this revised manuscript, the authors have made proper modification according to the reviewers' comments. I believe that this manuscript in the current form can be accepted for publication in Nature Communications.

Response: We thank the reviewer for recommending the publication of this paper.